

# Verifying triple and single Doppler lidar wind measurements with sonic anemometer data based on a new filter strategy for virtual tower measurements

Kevin Wolz[1], Christopher Holst[1], Frank Beyrich[2], Eileen Päschke[2], and Matthias Mauder[1,3]

[1]Institute of Meteorology and Climate Research - Atmospheric Environmental Research (IMK-IFU), Karlsruhe Institute of Technology (KIT), Garmisch-Partenkirchen, 82467, Germany
[2]Lindenberg Meteorological Observatory, German Meteorological Service (DWD), Tauche, 15848, Germany
[3]Institute of Hydrology and Meteorology, Technical University of Dresden, Dresden, 01069, Germany

*Correspondence to*: Kevin Wolz (kevin.wolz@kit.edu)

**Abstract.**

In this study, we compare the wind measurements of a virtual tower triple Doppler lidar setup to those of a sonic anemometer located at a height of 90 m above ground on an instrumented tower and with those of two single Doppler lidars to evaluate the effect of the horizontal homogeneity assumption used for single Doppler lidar applications on the measurement accuracy. The triple lidar setup was operated in a 90 m stare and a step/stare mode at six heights between 90 and 500 m above ground, while the single lidars were operated in a continuous scan Velocity-Azimuth-Display (VAD) mode where one of them had a zenith angle of 54.7 ° and the other one of 28.0 °. The instruments were set up at the boundary-layer field site of the German Meteorological Service (DWD) in July and August of 2020 during the FESST@MOL (Field Experiment on sub-mesoscale spatio-temporal variability at the Meteorological Observatory Lindenberg) 2020 campaign. Overall, we found good agreement of the lidar methods for the whole study period for different averaging times and scan modes compared to the sonic anemometer. For the step/stare mode wind speed measurements, the comparability between the triple lidar and the sonic anemometer was 0.47 m s$^{-1}$ at an average time of 30 minutes with a bias value of -0.34 m s$^{-1}$. For wind speed measured by one single lidar setup for the same period with an averaging time of 30 minutes, we found a comparability of 0.32 m s$^{-1}$ at an averaging time of 30 minutes and a bias value of -0.07 m s$^{-1}$ and values of 0.47 m s$^{-1}$ and -0.34 m s$^{-1}$ for the other one, respectively. We also compared the wind velocity measurements of the single and triple lidars at different heights and we found a decreasing agreement between them with increasing measurement height up to 495 m above ground for the single lidar systems. We found, that the single Doppler lidar with the increased zenith angle produced a poorer agreement with the triple Doppler lidar setup than the one with the lower zenith angle, especially at higher altitudes. At a height of 495 m above ground and with an averaging time of 30 minutes the comparability and bias for the larger zenith angle were 0.71 m s$^{-1}$ and -0.50 m s$^{-1}$, respectively, compared to values of 0.57 m s$^{-1}$ and -0.28 m s$^{-1}$ for the smaller zenith angle. Our results confirm that a single Doppler lidar provides reliable wind speed and direction data over heterogeneous but basically flat terrain in different



scan configurations. For the virtual tower scanning strategies, we developed a new filtering approach based on a Median
Absolute Deviation (MAD) filter combined with a relatively relaxed filtering criterion for the signal-to-noise-ratio output by
the instrument.

## 1 Introduction

Accurate knowledge of continuous temperature, humidity, and wind profiles in the lower atmosphere is indispensable for
enhancing our understanding of the water and energy cycles, particularly to account for land-atmosphere feedback processes
(Wulfmeyer et al., 2015; Wulfmeyer et al., 2018). Doppler lidars (DL) have shown to be a reliable tool for remotely-sensed
wind measurements of the lower atmosphere and have been successfully used in various studies e.g. for wind velocity,
turbulence, and aerosol concentration measurements (Sathe et al., 2011; Sathe and Mann, 2013; Emeis et al., 2007; Weitkamp,
2005; Ansmann and Müller, 2005). More recently, the benefits of the DL measurement technique have been harnessed in urban
areas where setting up towers is not feasible (Zeeman et al., 2022). The use of wind information from DL (in the data
assimilation process) can contribute to further increase the quality of the weather forecast (Pichugina et al., 2016; Illingworth
et al., 2019) and to provide further knowledge about site specific wind profile characteristics to improve the efficiency of wind
power plants (Frehlich and Kelley, 2008; Mariani et al., 2020). Compared to in-situ point wind measurement instruments, e.g.,
sonic anemometers, DLs have the advantage of providing temporally and spatially continuous wind measurements within the
atmospheric boundary layer. Aerosols act as distributed targets and scatter the transmitted laser pulses back to the DL where
they are detected by an optical detector. The line-of-sight (LOS) radial velocity is calculated with the help of the Doppler
frequency shift caused by the movement of aerosols through the laser beam (Huffaker and Hardesty, 1996). The result for each
DL beam is a vector of radial velocity data where each data point can be assigned to a certain distance from the DL system,
representing an averaged value over a range interval defined by the laser pulse duration.

To determine wind speed and direction from a single DL a set of different scanning methods are well-established (Werner,
2005). However, they all rely on the assumption of horizontal homogeneity of the wind field at the measurement height. The
three-dimensional wind components can be calculated from the radial velocity measurements of a single DL by using at least
three measurements taken with different linearly independent azimuth angles while choosing an appropriate averaging time
(Chanin et al., 1989). That can be achieved with multiple different scanning modes like the Doppler beam-swinging (Lundquist
et al., 2015), Velocity-Azimuth-Display VAD (Browning and Wexler, 1968), and six-beam methods (Sathe et al., 2015).

If, instead, three DL systems are used the three-dimensional wind vector can be calculated at every point where the beams of
the three systems intersect, provided that these beams are also linearly independent. The term virtual tower (VT) was used for
a similar scan pattern with only two DLs for the first time by Calhoun et al. (2004). By using three intersecting laser beams,
theoretically more precise measurements can be performed because the assumption of a homogeneous wind field at the
measurement height is not required in this case, which is particularly advantageous for measurements in areas with a





heterogeneous land cover or complex terrain. However, under the assumption of horizontal homogeneity, it is also possible to
calculate the wind vector at other heights where the beams do not meet.

Before even starting those measurements, the setup of the instruments already provides some possible error sources related to
the correct GPS location and north orientation of the instruments. Such position errors for a single DL will propagate when
measuring with multiple instruments and, therefore, have to be minimized as much as possible. Using a precise GPS device
and evaluating a hard target scan will help to reduce those errors. For a multi-DL set-up, also a correct time synchronization
between the systems is crucial, especially, when using fast-moving scan strategies. Otherwise, the systems will not measure at
the same location at the same time, which can lead to errors depending on the time delay and, mainly, the turbulence intensity
(i.e., similarity scales in time and space). To solve that problem various time synchronization programs are available, either
using GPS or a time server (Stawiarski et al., 2013).

The wind data retrieved from a single DL may still be of sufficient precision and accuracy for most applications, depending
on the site conditions. A thorough evaluation of such single DL retrievals requires a comparison with a collocated instrumented
tall tower as reference (Bonin et al., 2017), or a triple DL set-up, which does not require the above-mentioned homogeneity
assumption (Newman et al., 2016; Choukulkar et al., 2017). Newman et al. (2016) found an agreement in wind direction and
speed measurements between a single DL using the Doppler beam-swinging method and a triple DL setup of $r^2 = 0.99$ at a
height of 105 m above ground. Choukulkar et al. (2017) found uncertainties of between 0.50 m s$^{-1}$ and 1.0 m s$^{-1}$ for wind speed
and between 10 ° and 20 ° for wind direction comparisons of single and triple DL setups with a sonic anemometer depending
on the scanning strategies. Pauscher et al. (2016) pointed four distributed DL systems at a height of 188 m above ground where
they compared their wind measurements with those of a sonic anemometer mounted on a foldable boom at the mast there. For
their comparison of the radial velocity measurements of the four DLs with the projection of the sonic anemometer on the
beams of the respective DL they found root mean square differences (RMSD, also termed "comparability") between
0.079 m s$^{-1}$ and 0.116 m s$^{-1}$ and a value of $r^2$ between 0.858 and 1.00 for the four DL systems. Another interesting
intercomparison was conducted between an experimental bistatic DL, consisting of one transmitting unit and three receiving
units, and a sonic anemometer at a height of 30 m above ground in flat but heterogeneous terrain (Mauder et al., 2020). Their
results showed an agreement of the mean wind speed measurements and the standard deviations of the vertical wind speed
with a comparability of 0.082 m s$^{-1}$ and 0.020 m s$^{-1}$, respectively, and corresponding bias values for these two quantities of
0.003 m s$^{-1}$ and 0.012 m s$^{-1}$.

To investigate the influence of different zenith angles in single DL measurements, we compared data measured with two single
DLs operated in VAD scanning modes with the zenith angles of 54.7 ° and 28.0 ° to data obtained by a triple DL setup in VT
scan mode and by a sonic anemometer mounted at 90 m above ground on a 99 m tall, instrumented tower at the same site.
Moreover, relying on the comparison with in-situ data at a height of 90 m, we extend the single versus triple DL
intercomparison to measurement heights up to 500 m in order to assess the effect of the increase of the scan circle diameter of
a single DL operated in VAD mode vs. a "column reference". The measurements were taken in the northeast of Germany in
Falkenberg (Tauche) at the boundary-layer field site of the German Meteorological Service (DWD) in June, July, and August



2020. Our objective was to quantify the differences between single and triple DL measurements, how they compare to the sonic anemometer, their dependence on the averaging time and atmospheric conditions, and if we can see an influence of the

different zenith angles between both single DLs. We will provide uncertainty estimates for the tested retrieval methods and investigate their dependence on meteorological conditions, particularly atmospheric stability. We will present an analysis of the averaging times of 2, 10, and 30 minutes. Moreover, if a long-term deployment of DLs is intended, an adequate automated suite of data quality control tests needs to be developed and adapted to the specific instruments. Therefore, we will evaluate the use of different data filtering methods to improve the data quality of the triple DL setup. An overview over different data

filters commonly used for DL measurements is given by Beck and Kühn (2017).

In the following methods section, we describe the scan strategies we used in our triple and single DL setups and explain the difficulties we faced during the setup and alignment of the instruments. Then, we present the algorithm for calculating the horizontal wind speed and wind direction out of the radial velocities of the three individual DL systems and demonstrate the filtering criteria we used to filter out erroneous data. In the results section, we show the comparisons of triple DL with a sonic

anemometer at a height of 90 m above ground and with two single DLs at six different heights above ground, followed by a discussion of the results and our conclusions.

## 2 Methods

### 2.1 Set-up of the Doppler Lidar instruments

The data presented in this study was collected between 20 June and 10 August 2020 in Falkenberg (Tauche), Germany, during

the FESST@MOL (Field Experiment on sub-mesoscale spatio-temporal variability at the Meteorological Observatory Lindenberg) 2020 campaign, which was initiated by the Hans-Ertel-Center for Weather Research. The measurements took place at the boundary layer field site (GM) Falkenberg, Germany. During the campaign, a total number of eight DL systems were used with different scan strategies. Out of those eight, we used five DL systems for this study, two were operated by the DWD as single DLs in VAD mode, and the other three were set up as VT, operated by the KIT, one of which is shown in

Figure 1. VAD 1 was operated with a zenith angle of 54.7 ° and an azimuth resolution of between 1.5 ° and 4.5 °, whilst VAD 2 was running with a zenith angle of 28.0 ° and an azimuth resolution of between 33.0 ° and 36.0 °, both in a continuous scan mode. Further relevant details on the five systems are listed in Table 1. The distance between DL 1 and the vertical stare (VS) DL was 280.57 m, and the distance between DL 2 and VS DL was 282.72 m so that the three DLs formed an almost isosceles right-angled triangle (Figure 2). The time synchronization, which is crucial for combined DL settings, was enabled by using

the NetTime software. We used the network time protocol server 1.0 of the Technical University of Berlin, which was updated every 10 minutes and had a typical offset before the synchronization of ~ 35 ms. We deployed a Halo Photonics StreamLine XR as the VS DL and two Halo Photonics StreamLine systems as the scanning DLs (DL1 and DL2). A fourth and fifth scanning DL, two other Halo Photonics Streamline, were collocated nearby and operated in a VAD mode.





**Table 1: Technical details on the Halo Photonics Streamline DL systems.**

| Instrument | Serial number | Range Gate length (m) | Focus (m) | Pulses per ray | Pulse width (ns) | Pulse repetition frequency (kHz) |
|---|---|---|---|---|---|---|
| DL 1 | 0114-74 | 18 | 1000 | 15000 | 166 | 15 |
| DL 2 | 0114-75 | 18 | 1000 | 15000 | 169 | 10 |
| VS DL | 0319-161 | 18 | Inf | 15000 | 330 | 10 |
| VAD 1 | 0414-78 | 30 | 500 | 2000, 4000, 6000 | 180 | 10 |
| VAD 2 | 0120-177 | 30 | 2000 | 3000 | 180 | 10 |


When setting up the DL instruments for this study, we initially used a compass to orient them towards magnetic north, so that the instrument shows the azimuth angle in the unit degrees from north. The exact position of the DLs was determined with the help of an exact GPS tracker and the north orientation of the instruments was reconfirmed with the well-established hard targeting method as described for example by Rott et al. (2022). For the determination of the exact north orientation of the

instrument, we slowly moved the laser beam across the edge of a nearby building with previously determined GPS coordinates. The azimuth angle and the distance, at which the beam met the hard target, were determined from the backscatter coefficient in the data measured by the DL. This information was used to correct the azimuth angle of the DL after comparing that angle with the azimuth angle calculated from the GPS coordinates of the system and the obstacle. The distance between the DL systems, which is needed for the calculation of the wind variables, was calculated using GPS coordinates and basic

trigonometry.



**Figure 1: DL 2 with the 99 m tall lattice tower in the background, where a sonic anemometer is mounted at a height of 90 m above ground, on-site during the FESST@MOL 2020 campaign in Falkenberg (Tauche), Germany.**

## 2.2 Scan strategies

The three DLs for the triple DL setup were operated in a VT mode, which means that the beams of the DL systems are intersecting in a defined air volume at the same time to be able to calculate the wind vector. For our experiment, we evaluated two different VT scanning strategies, one, where the three DLs operate in stare mode pointing on one fixed point all the time, and a step/stare mode with several alternating intersection points. We operated one of the DLs (VS DL) in a continuous vertical stare mode and the other two instruments (DL 1 and DL 2) either in stare or in step/stare mode, where they scanned along the beam of the vertical stare DL and remained for 10 or 30 minutes at one of six measurement heights between 90 m and 500 m above this DL. The stare mode pattern only focused at a height of 90 m above the vertical stare DL to create a continuous data



set for a comparison with the sonic anemometer. The six heights of the step/stare mode were chosen to meet some of the respective range gates of the two DLs that were operated in VAD mode. These six heights were 95 m, 147 m, 199 m, 252 m, 304 m, and 494 m above the vertical stare DL. VAD 1 and VAD 2 were running in a continuous scan mode with high azimuth resolutions between 1.5 ° and 4.5 ° and between 33.0 ° and 36.0 ° and with zenith angles of 54.7 ° and 28.0 °, respectively. VAD scanning techniques use a fixed zenith angle while the azimuth angle is changing, meaning that the sensor head is rotating thereby resulting in a conical scan. The three-dimensional wind vector at a given height is derived from a sine fit through the data points representing the different azimuth positions. The continuous scan mode was chosen because both systems were operated to derive turbulent variables (using the methodology described in Smalikho and Banakh, 2017 - VAD 1) and wind gusts (see Steinheuer et al., 2022 - VAD 2) in addition to the mean wind vector. Both these scan modes require operation with a comparably low number of pulses per ray posing additional challenges concerning a suitable noise filtering (Päschke and Detring, 2023).

The total measurement period covered in this study was from 20 June until 10 August 2020. During this time, we conducted the 90 m stare measurements for six days until 25 June and then changed the scan schedule to the step/stare mode. In the period of the step/stare measurements, we lowered the time at which the beams stayed at each of the six heights from 30 to 10 minutes for the period of 23 July until 31 July. Therefore, we excluded this period for all the comparisons of the 30-minute averaged data. The periods where we conducted the different scanning strategies are also shown in Figure 3.

### 2.3 Calculation of the wind speed and direction from the Doppler Lidar data

A specific data processing chain was applied to calculate the three-dimensional wind vector. The main challenge after aligning the beams so that they all meet at a certain point is to select the correct range gate to ensure that the measurements at the same location are combined. Therefore, knowledge of the exact distance between the systems is crucial. We used the GPS coordinates and the Pythagorean theorem to calculate those distances. Then, we used basic trigonometry to calculate the heights above the VS DL at which the beams of each of the other two systems cross the beam of the VS DL using the elevation angle of each DL and its distance to the VS DL. With that information, the range gate in which the meeting point is located can be identified. In our case, which was very simple, since we only used a total of six different measurement heights, we chose the heights to either meet at the middle of a range gate or at the transition point of two consecutive range gates, in which case we averaged the radial velocity over both range gates. For the following steps, the respective radial velocities for each of the three DLs are used.

The radial velocity data of the three DLs were processed to calculate the three wind vector components u, v, and w via the transformation matrix (1),

$$\begin{bmatrix} u \\ v \\ w \end{bmatrix} = \begin{bmatrix} cos(\theta_1)\,sin(\varphi_1) & cos(\theta_1)\,cos(\varphi_1) & sin(\theta_1) \\ cos(\theta_2)\,sin(\varphi_2) & cos(\theta_2)\,cos(\varphi_2) & sin(\theta_2) \\ cos(\theta_3)\,sin(\varphi_3) & cos(\theta_3)\,cos(\varphi_3) & sin(\theta_3) \end{bmatrix}^{-1} * \begin{bmatrix} V_{r1} \\ V_{r2} \\ V_{r3} \end{bmatrix}, \tag{1}$$



where $V_r$ represents the radial velocity of the three different DL systems, $\theta$ the elevation angle, and $\varphi$ the azimuth angle of the corresponding DL system (Fuertes et al., 2014). We calculated the wind direction from the wind vector components by using trigonometry with the tangent of $u\,v^{-1}$. With this method, it is necessary to adjust the calculated wind direction to the correct

185  quadrant of the wind components, so that the wind direction is calculated as the direction where the wind comes from.

**2.4 Overview of the field site and data processing**

We tested different averaging times of 2, 10, and 30 minutes to study the dependence of the intercomparison results on the averaging time. To that end, we compared the DL data with data from a sonic anemometer mounted at the height of 90 m above ground on a 99 m tall tower located next to the VT measurements at the field site of the DWD in Falkenberg, Germany

190  (Figure 2). The area around the site can be characterized as flat with heterogeneous land cover on scales of a few hectometers to kilometers (Beyrich et al., 2006). The L-shaped field site was covered with short grass and the large fields surrounding the site from the southwest to the east of the tower were covered with maize. Figure 2 also shows the locations of the three VT DLs (DL 1, DL 2, VS DL) and the location of the two VAD DLs (VAD 1, VAD 2).





**Figure 2: Overview of the boundary layer field site (GM) Falkenberg for the intercomparison experiment, where the positions of the DLs and the lattice tower are marked with a scale of 1:2900. © Google Earth.**

For the comparison of the VT with the in-situ tower measurements, we only used the data when the three beams were intersecting at 90 m above the VS DL, meaning that only four hours of data were used every day for the step/stare mode. During the rest of the day, the VT was measuring at different heights, allowing for a comparison with the collocated single DLs that were operated in a VAD mode. For that second comparison, we used all data at the six selected measurement heights. The complete data processing was done using the programming language R (R version 3.6.3, RStudio version 2022.07.1+554, Boston, U.S.A). In Figure 3 we give an overview over the meteorological conditions as measured by the sonic anemometer during our observation period. During the whole period, the conditions were similar with maximum temperatures of around 25 °C to 35 °C and maximum wind speeds of around 7 m s$^{-1}$ to 10 m s$^{-1}$. The main wind direction was between 220 ° and 360 ° from north. There were a few days with lower temperatures and higher gust speeds, especially at the beginning of the observation period and at the beginning of July. Also, a steady rise of the temperature over the last days of the period is apparent.



**Figure 3: Overview over the meteorological conditions during the measurement period of 20 June 2020 – 10 August 2020 as measured by the sonic anemometer at a height of 90 m above ground on the lattice tower. We show temperature, wind speed, gust (a), and wind direction (b) with an averaging time of 30 minutes. We also added the periods where we used the different scanning strategies.**

## 2.5 Data Filtering

The sonic anemometer (USA-1, METEK, Elmshorn, Germany) is mounted 90 m above ground at a distance of 5 m from the tower on a boom fixed at the western side of the lattice tower and pointing towards 190 °. Therefore, the wind data of the sonic anemometer had to be filtered for wind directions from 0 ° to 50 ° from the north to avoid flow distortion by the tower despite the omnidirectional character of the instrument itself. Next to the sonic anemometer, at a distance of 25 cm from it, there is a LI7500 infrared gas analyzer mounted in the wind direction sector disturbed by the tower.

The data of the single DLs VAD 1 and VAD 2 are filtered with a modified version of the random sample consensus after Fischler and Bolles (1981). In that version first applied by Strauch et al. (1984) to radar wind profiler measurements the VAD



data are filtered using the consensus averaging method with the filter parameters $X = 3$ m s$^{-1}$ and 60 % for the smallest subset of data allowed.

We used different consecutive filtering methods for our VT DL measurements to exclude erroneous data and to assure high data quality. All of the following filters have been applied to the radial velocity data from the three DLs before combining

them to a VT for calculating the $u$, $v$, and $w$. Firstly, we remove the data with a high noise level by filtering with a relatively low Signal-to-Noise Ratio (SNR) + 1 threshold of 1.000. The SNR values are generally output by the instruments and serve as a quality indicator of the radial velocity measurements. As clearly visible in Figure 5a, a fair amount of erroneous data still remains after applying this filter. Secondly, we used the MAD filter, which had been originally defined by Mauder et al. (2013) for tower-based in-situ turbulence measurements. A similar approach is described as the interquartile filter described by

Hoaglin et al. (1983). Originally, we applied it to the time periods that we later averaged into for the comparisons with the other methods. Since that did not prove to be that effective (see Table 2) we decided to apply it to shorter time periods to increase the efficiency of the filter which is why the DL data were summarized into 30-second periods. If the data availability for a single period was below 30 % that 30-second period was excluded from further analysis since the following filter needs a minimum amount of data points to be applicable.

The MAD filter flags a data point $\chi_i$ as a spike, if it is outside the range of (2):

$$\langle \chi \rangle - \frac{q * MAD}{0.6745} \leq \chi_i \leq \langle \chi \rangle + \frac{q * MAD}{0.6745},$$   **(2)**

where $\langle \chi \rangle$ is the median of x, $MAD = \langle |x_i - \langle x \rangle| \rangle$ and $q$ is a threshold value that we set to 1 since it filtered out a sufficient amount of data that way. All spikes detected that way were discarded from further analysis. The MAD filter was originally used on 30-minute intervals and was altered for this purpose since we wanted it to be more sensitive to single erroneous values, and the 30-second intervals were found to be more reliable than the 30-minute intervals. The advantage when using the 30-

second test is that it does not directly discard the whole 30-minute period but, instead, only single 30-second periods. Thirdly, we applied another consistency test, where we checked if the difference in the radial velocity measurements of two consecutive range gates was higher than 1 m s$^{-1}$. If so, the radial velocity value of the higher range gate was discarded as well.

The final filter that we applied to the DL data is a test for unimodality of the frequency distribution of radial velocities during the 30 minutes averaging interval, which filters out periods where there are multiple peaks in the data (R-package "diptest" by

Martin Maechler, 2021). The test is based on Hartigan's dip statistic after Hartigan and Hartigan (1985), which checks if statistical data has more than one peak in its distribution by creating a unimodal distribution function that has the smallest value deviations from the empirical distribution function. By doing so, it creates the dip statistic using the largest of these deviations, which evaluates the probability of a bimodal distribution. This filter addresses the problem of noise around zero, which are erroneous radial velocity values around 0 m s$^{-1}$ that appeared mainly in higher range gates for DL 1 and DL 2. In

that case, the peak of the noise around 0 m s$^{-1}$ in the frequency distribution of the radial velocity values could be similar or even higher than the signal of the actual radial velocity. If that happens, the Median Absolute Deviation (MAD) filter (2) may



potentially filter out the correct data instead of the erroneous ones. We chose to apply this test on 30-minute time intervals since that way there is a sufficient amount of data points for the filter available. Also, it can detect periods where the MAD filter fails due to mainly erroneous values in the 30-second intervals. We did not find a period where the problem with noise around zero appears at the measurement heights we analyzed between 90 m and 500 m above ground. We, therefore, illustrate the problem of noise around zero in Figure 4 using data from a higher altitude (here 1220 m above ground). The red data points show the unfiltered data and the green points what remains after applying the SNR + 1 and MAD filters. The accumulation of green data points around 0 m s$^{-1}$ represents the erroneous data, whereas, the data around 10 m s$^{-1}$ represents the correct signal. In a situation like this, the dip statistic removes the whole 30-minute period.

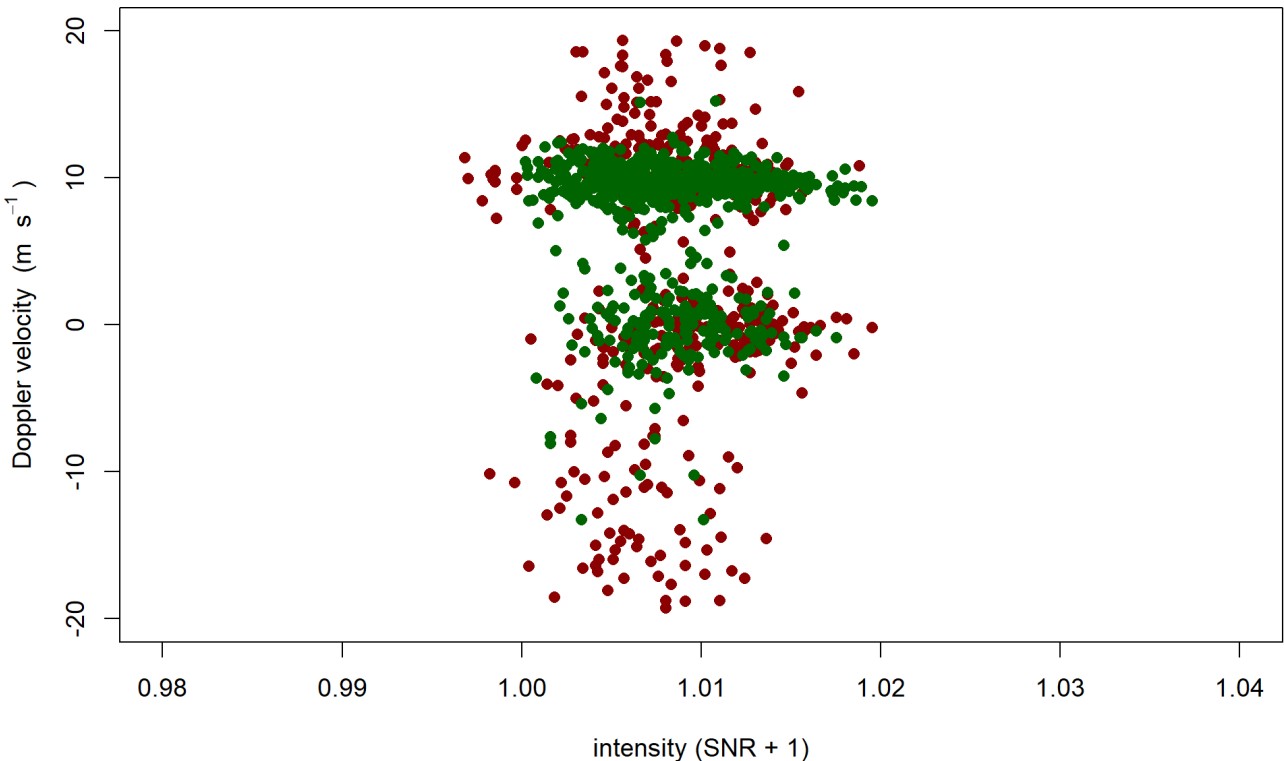

**Figure 4: Example plot to show the problem of noise around zero, the figure shows the unfiltered data (red) and the data after applying the SNR + 1 and MAD filters (green) for DL 2 for the period of 10:30 – 11.00 p.m. on 06 July 2020 for a single range gate at a height of 1220 m above ground.**

The MAD filter is only applicable for the step/stare and stare modes because the beam alignment has to be maintained at least for 30 seconds for the filter to be usable. When using a constantly moving scanning strategy like a range-height indicator (RHI) there would not be a sufficient amount of data points at each measurement height available to make the filter applicable. The





same goes for meteorological situations with sudden changes in wind speed or direction. When applying the MAD filter to constantly moving scanning strategies an increase of the time period the filter is used on would be necessary. If the actual radial velocity is around 0 m s$^{-1}$ the filter is not able to differentiate between correct data and noisy data around zero since the

frequency distribution of radial velocities will be unimodal in that case. The effect on the results, however, will be only minor. After applying all these filters, the remaining data are averaged further into 2-, 10-, and 30-minute periods. Nevertheless, only if there are more than 50 % of 30-second periods available for each averaging period meaning that in case of an averaging time of 1 minute, both 30-second periods had to be available. Table 2 shows the difference in the comparison between the sonic anemometer and the VT DL setup for an averaging time of 10 minutes when applying the filter to the radial velocity data of

each 10-minute or 30-second period. The main effect that can be seen is the increased number of remaining 10-minute periods. Besides that, the differences between the two methods shown here are only minor.

**Table 2: Difference for the comparison between sonic anemometer and VT measurements for the averaging time of 10 minutes when applying the MAD filter (2) to the radial velocity data of each 30-second and 10-minute period.**

| Filtering period | Mode | No. of observed periods | Measurement period (2020) | RMSD wind speed (m s$^{-1}$) | RMSD wind direction (°) | Pearson's r wind speed | Person's r wind direction |
|---|---|---|---|---|---|---|---|
| 30 sec | VT stare | 623 | 06-20 – 06-25 | 0.448 | 3.37 | 0.985 | 1.00 |
| 10 min | VT stare | 616 | 06-20 – 06-25 | 0.430 | 3.24 | 0.983 | 1.00 |
| 30 sec | VT step/stare | 795 | 06-26 – 08-10 | 0.521 | 5.91 | 0.973 | 0.998 |
| 10 min | VT step/stare | 790 | 06-26 – 08-10 | 0.510 | 5.09 | 0.972 | 0.999 |

The efficiency of the data filters and the number of remaining data points are shown in Table 3 as an example of the data with an averaging time of 1 min for the triple DL stare mode compared to the sonic anemometer data at a height of 90 m above ground, by applying either none or all the filters during the processing steps. It does not show a clear effect of the filtering methods, the RMSD and Pearson's *r* values stay fairly similar during the filtering process which might be since the VT DL setup already compares well with the sonic anemometer data at the height of 90 m above the ground even without applying

any filtering methods. The main effect of the filtering can be seen by the difference in the RMSD coefficients for the horizontal wind speed for the step/stare mode from 0.888 m s$^{-1}$ to 0.642 m s$^{-1}$.





**Table 3: Difference of the comparison for the VT DL setup with the sonic anemometer at a height of 90 m above ground for the averaging time of 1 minute and the two different scan strategies between the filtered and unfiltered data. The statistical parameters used for the comparison are the RMSD and Pearson's r coefficient.**

| Mode (VT) | Filtered (yes/no) | Measurement period (2020) | Number of observed periods | RMSD wind speed (m s$^{-1}$) | RMSD wind direction (°) | Pearson's $r$ wind speed | Pearson's $r$ wind direction |
|---|---|---|---|---|---|---|---|
| stare | no | 06-20 – 06-25 | 6600 | 0.802 | 5.06 | 0.928 | 0.999 |
| stare | yes | 06-20 – 06-25 | 6445 | 0.795 | 5.24 | 0.934 | 0.999 |
| step/stare | no | 06-26 – 08-10 | 8929 | 0.888 | 11.7 | 0.915 | 0.990 |
| step/stare | yes | 06-26 – 08-10 | 8568 | 0.642 | 10.58 | 0.958 | 0.992 |

To demonstrate the effect of the filtering described above on the data availability, we compared the radial velocity with the SNR + 1 values in a similar way as in the study of Päschke et al. (2015). We restricted this analysis to the heights of 60 m – 500 m above ground since this is the range that we used for our further analysis. The lowest two range gates of the observations were discarded regardless since they tend to be erroneous for this instrument. As an example, we show the unfiltered data in a) and the data filtered with the SNR + 1 filter and the MAD filter in b) for DL1 on 14 July 2020 in Figure 5. The SNR filter removes all the data with an SNR + 1 value lower than 1 (dark green line in a). The rest of the removed data got filtered out by the MAD filter. The range of radial velocity values for the Halo Photonics DL is from - 19.4 m s$^{-1}$ up to 19.4 m s$^{-1}$ for radial velocity. The poor-quality data is, therefore, spread out over this entire range. Figure 5 also shows the problem of working only with an SNR filter alone: To remove all erroneous values spreading over the whole range of the radial velocity data in a) we would have had to choose a relatively high SNR threshold of around 1.020 (red line). That would in turn lead to a big data loss of usable data in the center of the figure. In contrast, using a low SNR + 1 threshold of 1.0 together with the MAD filter described above also removes the poor-quality data but more of the high-quality data remains. Therefore, we chose this filtering procedure together with the dip test for the results presented below.



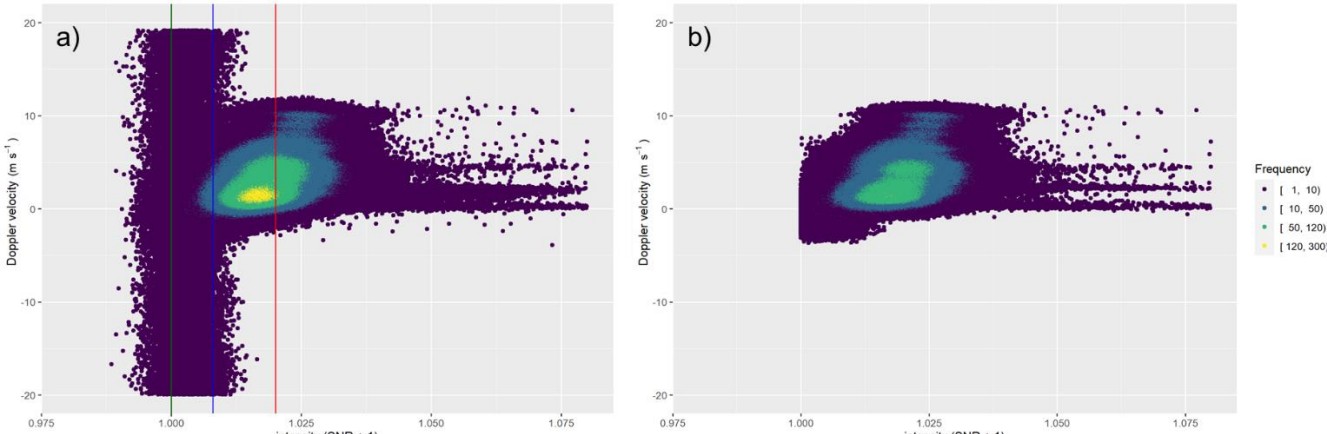

**Figure 5: Comparison between unfiltered (a) and filtered (b) data of DL 1 as an example on 14 July 2020 on heights of 60 m – 500 m above ground. The colored lines in a) mark the different SNR + 1 thresholds of 1.000 (dark green), 1.008 (blue), and 1.020 (red).**

## 3 Results

To illustrate the time-spatial availability of the data used for this study, Figure 6 shows the results of VAD 1, triple DL, and sonic anemometer measurements as an example for 06 July 2020 with an averaging time of 10 minutes. VT measurements are evenly distributed over the entire day with an ascending measurement height of intersecting laser beams before a new cycle of ascending VT measurements begins. In contrast, VAD measurements show a good availability during daytime at many heights simultaneously up to 500 m, while the data are limited to the lower heights during the nighttime periods. Overall, the VAD measurements cover a much larger space in the height-time domain than the VT measurements. Although, under the assumption of horizontal homogeneity the measurements of the three DLs of the VT setup could also be used to calculate the horizontal wind speed at other heights than those where the beams meet. However, because of the relatively low elevation angles of the two slanted DL beams the horizontal distance of the scattering volumes rapidly increases apart from the intersection point, especially at low intersection heights.



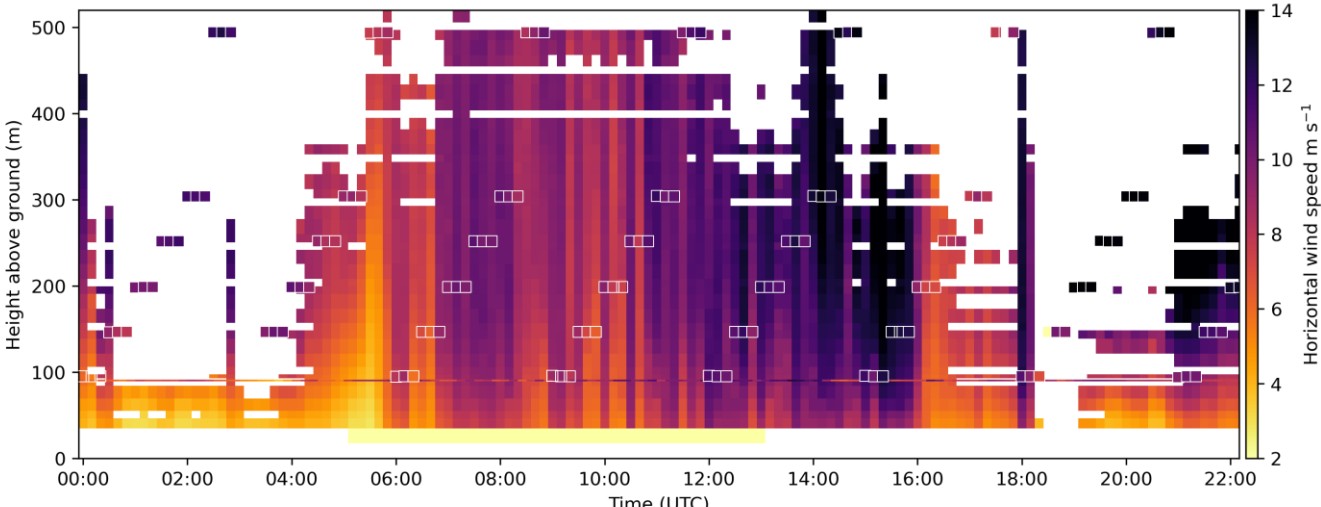

**Figure 6: Overview of the different ten minutes mean horizontal wind speed measurement methods we used as an example for 06 July 2020. The background data are the results of the VAD 1 measurements, the white boxes show the results of the VT DL step/stare mode at the six different heights. The line at the height of 90 m above ground shows the sonic anemometer data.**

For both VT scan modes and the VAD DL measurements, we performed comparisons with the sonic anemometer. To that end, we conducted a regression analysis and calculated the RMSD, which is also called comparability, through (3)

$$RMSD = \sqrt{\frac{\sum_{t=1}^{T}(\bar{x} - x_t)^2}{T}},$$

(3)

where $\bar{x}$ represents the mean value of the desired variable and $x_t$ the individual values. We also calculated the bias value, which is the mean value of the difference between sonic anemometer and DL measurements. These measures, comparability, and bias have also been used to characterize the uncertainty of turbulence measurements based on sonic anemometers intercomparison measurements in the past (e.g. Mauder et al., 2007; Mauder and Zeeman, 2018) and also for comparisons between DL and sonic anemometer (Mauder et al., 2020).

The comparisons were performed for the three different averaging times of 2, 10, and 30 minutes. Figure 7 shows the results for the VT DL 90 m stare measurements with an averaging time of 30 minutes. The statistical results for the comparisons between DLs and sonic anemometer, including the different averaging times and the different scan strategies, can be found in Table 4 and Table 5. The wind direction data of VAD 2 could not be used since there were some internal issues with that particular DL preventing the correct measurement of the wind direction.



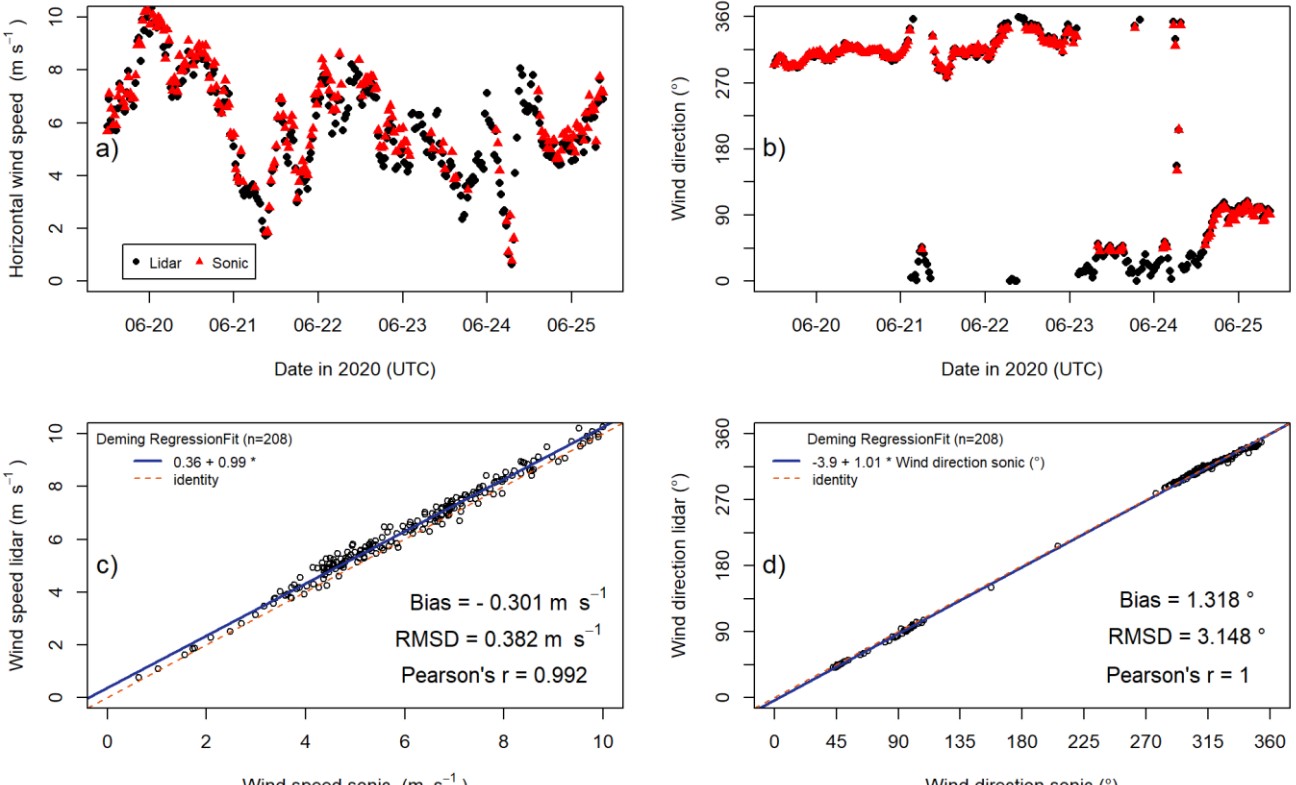

**Figure 7: Comparison between the triple DL VT for the 90 m stare mode with the sonic anemometer at a height of 90 m above ground during the period between 20 June – 25 June 2020. The 30 min mean horizontal wind speed (a) and wind direction (b) are compared with the respective sonic anemometer measurements. The linear regression analysis for the horizontal wind speed (c) and**
**wind direction (d) are also shown, including Pearson's *r*, RMSD coefficients, and the bias values.**

Figure 7 shows that the overall agreement between wind speed and direction data from the triple DL VT and sonic anemometer is fairly good. Especially, the Pearson coefficients, the RMSD, and the bias values for wind direction and mean horizontal wind speed further support the data seen in the plots. They reach Pearson coefficients of 0.992 and 1.00 for the horizontal wind speed and direction, respectively. The corresponding RMSD values are 0.382 m s$^{-1}$ and 3.148 ° and bias values of -0.301 m s$^{-1}$

and 1.318 °. The application of linear statistics to cyclical variables like the wind direction in this case requires certain caution with interpretation. The systematic agreement between the independently measured and derived quantities is good, despite notable differences in sampling volumes, time resolutions and measurement principles. This confirms that instrument alignment and setup has been carried out with due diligence.

Table 4 and Table 5 show that the agreement between triple DL VT, single VTs, and sonic anemometer decreases over all

measurements when the averaging time decreases, for example, when comparing the above-mentioned values from Figure 7 with those of an averaging time of 2 min, down to Pearson's r coefficients of 0.955 for the horizontal wind speed and 0.999 for the wind direction. The respective RMSD values are 0.639 m s$^{-1}$ and 4.20 ° and the bias values are -0.270m s$^{-1}$ and 1.33 °.



No big difference can be seen for the comparisons with the sonic anemometer between both measurement modes, the RMSD wind direction values show a slightly better agreement with the anemometer for the stare mode. This is reasonable since the second mode at a given height (90 m above ground) is effectively a stare mode as well. However, the database and also the meteorological conditions were different for the two comparison periods. Again, compared to the 30 min data from Figure 7, the corresponding 30 min values for the step/stare mode for the mean horizontal wind speed and direction, respectively, are 0.981 and 0.999 for the Pearson coefficient, 0.450 m s$^{-1}$ and 4.58 ° as RMSD values, and -0.164 m s$^{-1}$ and 2.21 ° as bias values. For the horizontal wind speed comparison, the RMSD values of the VAD 1 measurements values are slightly lower than for the VT measurements while the RMSD values for the wind direction are similar between both methods. The Pearson *r* coefficient showed comparable values for the horizontal wind speed comparison and wind direction data between the VT stare and the VAD measurements. For the step/stare mode, it showed a better agreement for the VAD measurements. Besides that, it is apparent that the agreement is higher between VAD 1 and the sonic anemometer than between VAD 2 and the sonic anemometer. For example, for the period of the VT stare mode and an averaging time of 2 minutes, VAD 1 has a horizontal wind speed comparability of 0.521 m s$^{-1}$ and a bias of -0.154 m s$^{-1}$, whereas, VAD 2 reached a comparability of 0.669 m s$^{-1}$ and a bias of -0.412 m s$^{-1}$.



**Table 4: Results of the comparisons for the VT stare mode of the triple DL setup and the VAD mode of the single DLs with the sonic**
**anemometer at a height of 90 m above ground for the different averaging times of 2, 10, and 30 minutes in the time frame of 20 June**
**- 25 June 2020 for time periods where data was available for all methods.**

| Mode | Averaging time (min) | Number of observed periods | RMSD wind speed (m s⁻¹) | RMSD wind direction (°) | Pearson's $r$ wind speed | Person's $r$ wind direction | Bias wind speed (m s⁻¹) | Bias Wind direction (°) |
|---|---|---|---|---|---|---|---|---|
| VT stare | 2 | 2885 | 0.639 | 4.20 | 0.955 | 0.999 | -0.270 | 1.33 |
| VAD 1 | 2 | 2885 | 0.521 | 4.10 | 0.967 | 0.999 | -0.154 | 1.23 |
| VAD 2 | 2 | 2885 | 0.669 | / | 0.962 | / | -0.412 | / |
| VT stare | 10 | 618 | 0.447 | 3.38 | 0.985 | 1.00 | -0.300 | 0.664 |
| VAD 1 | 10 | 618 | 0.340 | 3.38 | 0.988 | 1.00 | -0.171 | 0.884 |
| VAD 2 | 10 | 618 | 0.613 | / | 0.986 | / | -0.525 | / |
| VT stare | 30 | 208 | 0.382 | 3.15 | 0.992 | 1.00 | -0.301 | 1.32 |
| VAD 1 | 30 | 208 | 0.340 | 3.24 | 0.993 | 1.00 | -0.261 | 1.49 |
| VAD 2 | 30 | 208 | 0.636 | / | 0.993 | / | -0.589 | / |





**Table 5: Results of the comparisons for the VT step/stare mode of the triple DL setup and the VAD modes of the single DLs with the sonic anemometer at a height of 90 m above ground for the different averaging times of 2, 10, and 30 minutes in the time frame of 26 June - 10 August 2020 for time periods where data was available for all methods.**

| Mode | Averaging time (min) | Number of observed periods | RMSD wind speed (m s⁻¹) | RMSD wind direction (°) | Pearson's $r$ wind speed | Person's $r$ wind direction | Bias wind speed (m s⁻¹) | Bias Wind direction (°) |
|---|---|---|---|---|---|---|---|---|
| VT step/stare | 2 | 1686 | 0.547 | 4.43 | 0.970 | 0.998 | -0.102 | 1.20 |
| VAD 1 | 2 | 1686 | 0.437 | 3.98 | 0.979 | 0.998 | 0.022 | 0.413 |
| VAD 2 | 2 | 1686 | 0.501 | / | 0.977 | / | -0.242 | / |
| VT step/stare | 10 | 799 | 0.504 | 5.09 | 0.972 | 0.999 | -0.172 | 2.09 |
| VAD 1 | 10 | 799 | 0.333 | 5.09 | 0.985 | 0.999 | -0.010 | 2.09 |
| VAD 2 | 10 | 799 | 0.453 | / | 0.983 | / | -0.299 | / |
| VT step/stare | 30 | 217 | 0.450 | 4.58 | 0.981 | 0.999 | -0.164 | 2.21 |
| VAD 1 | 30 | 217 | 0.323 | 5.67 | 0.988 | 0.999 | -0.066 | 2.23 |
| VAD 2 | 30 | 217 | 0.468 | / | 0.986 | / | -0.336 | / |

The results for the comparison between the triple DL VT step/stare measurements and the VAD 1 and VAD 2 measurements
are shown in Table 6 and Table 7 for the same time periods, respectively. We compared these measurements at the six different heights of the step/stare mode for their mean horizontal wind speed values by analyzing the RMSD and bias values between both measurements. Additionally, we also added the number of data points used in the comparison, Pearson's $r$-value, the RMSD, and the bias value for the whole comparison. It has to be noted that we could not measure at all six heights at the same time since the beams of the VT DL setup were always positioned at one fixed height. The values show a clear tendency towards
a poorer agreement with increasing height above ground for both VAD DLs. They also show that the absolute agreement between both methods decreases with shorter averaging times. The comparability and bias of VAD 2 and VT DL systems are better compared to those of VAD 1 and VT DL, especially at higher altitudes. For example, on a height of 494 m above ground and with an averaging time of 2 minutes VAD 1 reaches values of 1.02 m s⁻¹ for the comparability and a bias of -0.693 m s⁻¹, whereas, VAD 2 reaches values of 0.786 m s⁻¹ and -0.406 m s⁻¹, respectively.




**Table 6: Medians of the mean horizontal wind speed relative differences of VAD 1 - VT DL setups for the three different averaging times of 2, 10, and 30 minutes at the six different heights above ground used in the VT step/stare mode in the period of 26 June – 10 August 2020. The table also includes the mean horizontal wind speed measured by the VT DL setup, RMSD, and bias values for the averaging time of 30 minutes.**

| Height above ground (m) | RMSD 2-min (m s⁻¹) | Bias 2-min (m s⁻¹) | RMSD 10-min (m s⁻¹) | Bias 10-min (m s⁻¹) | RMSD 30-min (m s⁻¹) | Bias 30-min (m s⁻¹) | Wind speed 30-min (m s⁻¹) |
|---|---|---|---|---|---|---|---|
| 95 | 0.369 | -0.128 | 0.363 | -0.156 | 0.276 | -0.091 | 4.78 |
| 147 | 0.456 | -0.250 | 0.397 | -0.281 | 0.385 | -0.207 | 5.34 |
| 199 | 0.424 | -0.174 | 0.416 | -0.270 | 0.375 | -0.195 | 5.63 |
| 252 | 0.459 | -0.183 | 0.403 | -0.266 | 0.460 | -0.193 | 5.78 |
| 304 | 0.518 | -0.219 | 0.489 | -0.303 | 0.462 | -0.206 | 5.81 |
| 494 | 1.02 | -0.693 | 0.924 | -0.684 | 0.707 | -0.500 | 5.14 |
| Data points (total) | 10787 | | 5072 | | 1351 | | |
| Pearson's *r* (total) | 0.985 | | 0.988 | | 0.989 | | |
| RMSD (total in m s⁻¹) | 0.544 | | 0.512 | | 0.452 | | |
| Bias (total in m s⁻¹) | -0.250 | | -0.313 | | -0.223 | | |




**Table 7: Medians of the mean horizontal wind speed relative differences of VAD 2 - VT DL setups for the three different averaging times of 2, 10, and 30 minutes at the six different heights above ground used in the VT step/stare mode in the period of 26 June – 10 August 2020. The table also includes the mean horizontal wind speed measured by the VT DL setup, RMSD, and bias values for the averaging time of 30 minutes.**

| Height above ground (m) | RMSD 2-min (m s⁻¹) | Bias 2-min (m s⁻¹) | RMSD 10-min (m s⁻¹) | Bias 10-min (m s⁻¹) | RMSD 30-min (m s⁻¹) | Bias 30-min (m s⁻¹) | Horizontal wind 30-min (m s⁻¹) |
|---|---|---|---|---|---|---|---|
| 95 | 0.352 | 0.136 | 0.378 | 0.132 | 0.373 | 0.177 | 4.78 |
| 147 | 0.352 | 0.016 | 0.322 | 0.054 | 0.385 | 0.097 | 5.34 |
| 199 | 0.378 | 0.040 | 0.379 | 0.025 | 0.407 | 0.064 | 5.63 |
| 252 | 0.423 | -0.023 | 0.390 | -0.046 | 0.468 | -0.005 | 5.78 |
| 304 | 0.475 | -0.010 | 0.441 | -0.043 | 0.479 | -0.002 | 5.81 |
| 494 | 0.786 | -0.406 | 0.692 | -0.377 | 0.571 | -0.279 | 5.14 |
| Data points (total) | 10787 | | 5072 | | 1351 | | |
| Pearson's $r$ (total) | 0.987 | | 0.987 | | 0.988 | | |
| RMSD (total in m s⁻¹) | 0.458 | | 0.436 | | 0.447 | | |
| Bias (total in m s⁻¹) | -0.020 | | -0.029 | | 0.021 | | |

Next, we investigate potential factors that might have affected the measurement quality of the triple and the single DL measurements, respectively. To that end, we show the difference between the VT 30-minute mean horizontal wind speed measurements from the sonic anemometer in Figure 8 and the same difference for the DL 1 data from the sonic anemometer in Figure 9. We evaluate a potential influence of the time of the day, the atmospheric stability expressed by the Obukhov-length $L$, the wind direction, and the mean wind speed. We took all of that information from the sonic anemometer measurements. The comparison with the atmospheric stability is motivated to see if there is especially an influence of unstable conditions visible on the measurements of the single DL. Since the horizontal homogeneity assumption might not be met anymore if the atmospheric conditions are unstable, which leads to an increase in turbulence. Both figures show similar behavior, and at least at this height of 90 m above ground, there is no strong influence of stability and time of the day visible. However, a sinus-wave-like behavior can be recognized in both figures as a function of the wind direction (Figure 8c and




Figure 9c). Figure 10 shows the difference between the triple DL VT and the VAD 1 setups as a function of atmospheric stability and mean wind speed, but no clear dependence is visible. The figure again shows a tendency for larger differences between both methods at higher altitudes. The data points in these two plots do not completely align as we had to limit the range of the x-axis to be able to show the vast majority of the data adequately.

**Figure 8: Potential factors that might affect the measurement quality of the triple DL VT step/stare measurements are shown for the period 26 June 2020 - 10 August 2020. The y-axis shows the difference in the 30 min average horizontal wind speed measurements between the sonic anemometer and VT DL at the height of 90 m above ground. These data are compared with (a) the time of the day, with the 30 min mean values shown as the red line, (b) the Obukhov length L, (c) the wind direction measurement of the sonic anemometer, and (d) the mean horizontal wind speed measurement of the sonic anemometer.**





**Figure 9: Potential factors that might affect the measurement quality of the VAD 1 measurements are shown for the period 26 June 2020 - 10 August 2020 for the same time slots as in Figure 8. The y-axis shows the difference in the 30 min average horizontal wind speed measurements between the sonic anemometer and VAD 1 at the height of 90 m above ground. These data are compared with (a) the time of the day, with the 30 min mean values shown as the red line, (b) the Obukhov length L, (c) the wind direction measurement of the sonic anemometer, and (d) the mean horizontal wind speed measurement of the sonic anemometer.**



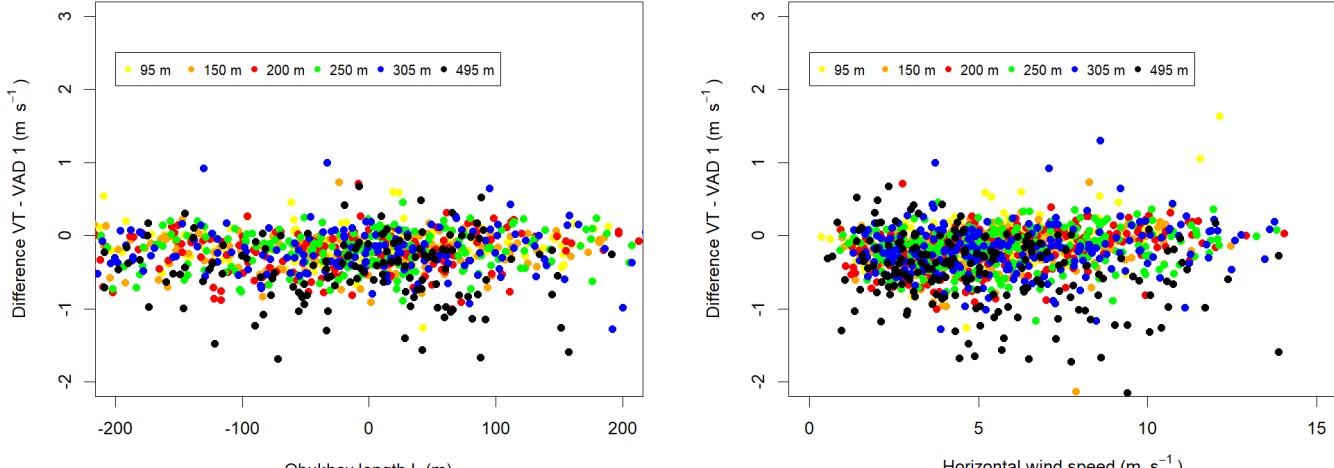

**Figure 10: Potential factors that might affect the difference between VT and VAD 1 measurements depending on the height above**
**ground (different colors) are shown for the period 26 June 2020 – 10 August 2020. The y-axis shows the difference in the 30 min**
**average horizontal wind speed measurements between VT DL and VAD 1 at the six different measurement heights of the step/stare**
**setup. These data are compared with (a) the Obukhov length L and (b) the horizontal wind speed measured by the VT DL method.**

## 4 Discussion

We designed this data processing method with the aim to calculate horizontal wind speed and direction from VT measurements
in an efficient and effective way. The method mainly uses basic trigonometry and is, therefore, easy to apply. Most importantly,
the correct orientation and alignment besides regular checks of the instrument's status during the campaign are critical for
successful measurements. The 90 m VT stare -measurements show a better agreement between DL and sonic anemometer than
the step/stare measurements. Since we only used the 90 m data of the step/stare measurements for the comparisons this
difference in comparability was not expected. The fact that the scanner measures at each height over 30 minutes makes it
effectively the same as the 90 m stare for that period. The stare measurements were collected over a period of only six days,
whereas the step/stare measurements were conducted over a period of 44 days. The mean horizontal wind speed measured by
the sonic anemometer for the two observation periods was 6.41 m s$^{-1}$ for the stare mode and 5.00 m s$^{-1}$ for the step/stare mode.
The (much longer) second period covered a larger variety of weather situations, including more weak-wind situations, which
might explain the differences in the comparison results. However, when investigating different potential sources of influence
like atmospheric stability we could not demonstrate any dependency. In total, our dataset covers 50 days and can therefore be
considered as a sufficiently long period to verify the DL measurements with the sonic data.

It must be noted that we were not fully confident in the measurements of the DL 1 system for higher range gates since the data
quality of the measurements of that particular DL deteriorated faster with increasing height than for the other systems, which
is why we introduced the filter that checked for the difference in radial velocity measurements of consecutive range gates. We,
therefore, assume that the results of the VT measurements could be improved further by using DL systems with a better signal-



to-noise ratio. The poorer performance of that DL 1 system resulted in a higher amount of data that was discarded by our filtering algorithm, especially at higher altitudes. We had to choose relatively relaxed filter criteria as a compromise between securing data quality while still maintaining a sufficiently large sample size.

The filters did not show a large influence on the comparability with the sonic anemometer, since the unfiltered data already agreed well at this relatively low height of 90 m above ground, which is the only height at which we could compare the VT setups with a sonic anemometer and major issues with a higher amount of erroneous data only occurs at higher altitudes. This leaves little room for improvement for the data filters and might, therefore, limit their effectiveness. Nevertheless, we were able to get some further insight into the data filters by looking at the SNR + 1 together with the radial velocity values. The difference between filtered and unfiltered data is visible in those plots as we showed in Figure 5 for 14 July 2020 as an example

day. We also showed that our data filters can handle difficult situations, although at the cost of losing a 30-minute time period in situations like this. In our VS DL, we found strange behavior in 2/3 of the observed days. As shown in Figure 11, in those cases the VS DL sometimes shows radial velocity values at around 10 m s$^{-1}$. We could not find an explanation for this behavior, but the data filters were able to remove those data effectively.

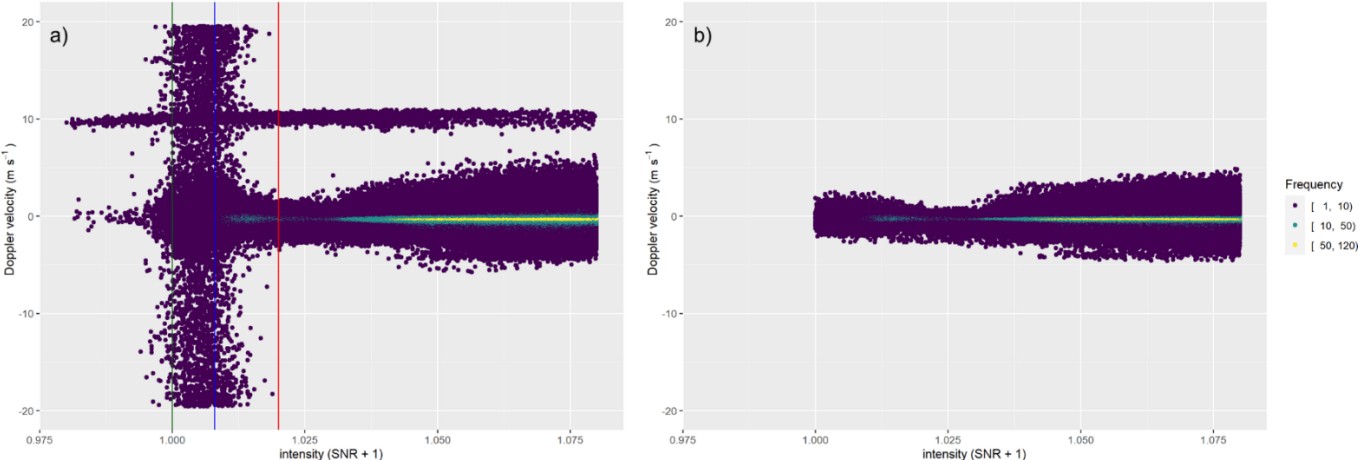

**Figure 11: Comparison between unfiltered (a) and filtered (b) data of VS DL on 06 July 2020 at the heights of 60 m – 500 m above ground. The colored lines in a) mark the different SNR + 1 thresholds of 1.000 (dark green), 1.008 (blue), and 1.020 (red).**

The good agreement between VT measurements and sonic anemometer measurements lends additional confidence to the triple DL measurements so that they can serve as a reference for the comparison between the VT measurements and the single DL VAD measurements at higher altitudes for which no in-situ data are available.

Using only one DL has the advantage that the setup of the instrument is simplified. Multiple DLs also represent multiple error sources, whilst a single DL reduces the risk of malfunctions since only one instrument must be operated. Moreover, the setup of the instruments presents another error source when using multiple DLs compared to a single instrument, since the north orientation has to be carefully verified for all three systems. Hence, single DLs are certainly easier to handle and more cost-efficient than using a triple DL setup. This leaves the question of whether the slightly increased data quality of the VT





measurements and the better range at nighttime is worth maintaining multiple systems compared to a single DL at a slightly poorer data quality but lower cost. Moreover, the nocturnal range of single DL measurements can potentially be further increased by increasing the number of pulses per ray used for the VAD measurements. It also has to be mentioned that the scan modes for VAD 1 and VAD 2 were mainly chosen to measure turbulent variables and wind gusts and are, therefore, not optimal for the measurement of the mean horizontal wind speed. Choosing another VAD configuration for the single DLs

should further improve their performance.

The VT step/stare is especially useful when a measurement at one or multiple certain heights is required. It gives the possibility to focus the beams at those heights for the same time as the later used averaging period. We demonstrated that the VAD DLs have the advantage of being able of providing continuous profiles of wind data for larger heights above ground. To achieve the same height profile using a VT mode another measurement regime using for example an RHI scan regime is required. Such

a method, where the beams of the two outer DL systems are constantly moving, would also provide a wind profile between the two chosen start- and end points when using a temporal average. However, this method requires an exact temporal synchronization of the instruments to guarantee that all three instruments measure at the same location at the same time. Moreover, the number of radial velocity measurements per height during each averaging interval is much lower for such a VT using RHI scans compared to a VAD or a step stare mode.

At least for the relatively low height of 90 m above ground, both methods, VT and VAD, show a good comparability with the sonic anemometer data. Almost no difference can be seen between the VT and VAD mean horizontal wind speed data. The VAD measurements show a slightly worse comparability when compared to the VT stare wind direction measurements. Since a single DL performs a conical scan to allow for the calculation of the three-dimensional wind vector, the radius of that circle increases with increasing height above ground. The corresponding averaging area, therefore, also increases with increasing

height. In comparison, the VT setup can focus on a single point at each height above the VS DL, so that we expected an increasing deficit in data quality of the single DLs with increasing heights due to a possible violation of the underlying homogeneity assumption. Considering the good statistical agreement for the horizontal mean wind in this intercomparison experiment, the VAD DLs seem to perform equally well for wind measurements at our site with only small deficits compared to the VT data.

We have found that the agreement between the VT and single DL measurements decreases with increasing height above ground and with decreasing averaging time. However, both methods performed equally well at the height of 90 m above ground when compared to the sonic anemometer. This further strengthens our confidence in the VT step/stare measurements for, especially, longer averaging times, where it benefits from the longer fixed position of the intersecting beams at a certain height. We could not find any dependence on the atmospheric stability or wind speed which could have potentially led to the higher differences

between VT and single DL measurements at higher altitudes. The agreement of VAD 1 with the sonic anemometer is higher than the agreement of VAD 2. On the contrary, the reverse is observed when comparing them to the VT setup, where VAD 2 agrees better. The main difference between VAD 1 and VAD 2 is the different zenith angles of 54.7 ° and 28.0 ° used for the VAD modes, respectively. This indicates that smaller zenith angles lead to more accurate results, especially in higher altitudes,



in comparison to larger zenith angles. This is as expected since the size of the scan circle gets larger when a larger zenith angle is used.

In comparison to the data of the VAD measurements, the values of the VT step/stare setup wind measurements show slightly lower values for the mean horizontal wind speed over all heights and averaging times. Remembering our argument from before based on the with height increasing spatial averaging area for the VAD measurements we are confident that the VT measurements present the actual state of the wind field at the VT location more precisely and are, therefore, more trustworthy at higher altitudes. This means that the measurements of the single DL slightly overestimate the horizontal wind speed. Nevertheless, the difference also at larger measurement heights between the two methods is still acceptable leading to the conclusion that a single DL can be used reliably for sites with flat terrain with heterogeneous land cover, similar to the Falkenberg test site. This holds especially true if one is mostly interested in lower measurement heights and if nighttime data at greater heights are not of critical interest.

Furthermore, we found that both methods, VT and VAD, perform very similarly independent of the different potential influencing factors, such as atmospheric stability, time of the day, and wind speed. However, a dependence on wind direction of the agreement for both DL configurations can be seen, which may point to issues with the wind direction measurement of the sonic anemometer, which was not known before. The wind speed differences as a function of wind direction exhibit a sinus-wave-like deviation curve for both systems, which might be related to the position of the sonic anemometer on the tower. Although we filtered the sonic anemometer data for directions between 0 and 50 °, our data indicate that there might be a possible distortion of the sonic data also for the wind direction sector around 210 °. This might be due to either the mounting of the gas analyzer close to the sonic anemometer or a tower-induced modification of the upstream flow.

Rahlves et al. (2022) found in their study, simulating different VAD scan strategies in large-eddy simulations and comparing different averaging times, that the DL measurements worked best in more stable atmospheric conditions. On the contrary, our data do not indicate a stability dependence. They also found that the performance of the DL measurements decreases when lower averaging times are aggregated. This finding is in agreement with our data. Park et al. (2018), who compared single DL measurements operated in a VAD mode and a Doppler-Beam-Swinging method for different averaging times between 1 minute and 15 minutes with simultaneous radiosonde soundings while choosing a range gate length of 75 m, also saw a better performance of the DL systems when the averaging time increased. Robey and Lundquist (2022) simulated a virtual DL in an idealized large-eddy simulation using a Doppler-Beam-Swinging method and found the strongest error in strong convective conditions and at larger heights above ground. Our results also compare well to the findings of other intercomparison experiments between DL and Sonic anemometers (Newman et al., 2016; Pauscher et al., 2016; Mauder et al., 2020; Choukulkar et al., 2017), who show a very good agreement for wind speed and direction measurements with a comparability as good as in intercomparison experiments between different types of sonic anemometers.



## 5 Summary and Conclusions

In this study, we compared the measurements of a VT triple DL setup with those of two VAD single DL setups and with a sonic anemometer mounted at 90 m above ground on a lattice tower. We collected the data for 52 days in the summer of 2020. Our results indicate good comparability between those three methods for the observation level of 90 m. The differences between the triple and single DL setups are small when both systems report data of high quality, which lends confidence in the single DL setups despite its reliance on the assumption of horizontal homogeneity. The comparabilities for horizontal wind speed between the sonic anemometer and the triple DL were 0.382 m s$^{-1}$ for the stare mode and 0.450 m s$^{-1}$ for the step/stare scan mode at an averaging time of 30 minutes. The corresponding bias values were 0.301 m s$^{-1}$ and -0.164 m s$^{-1}$, respectively. For the VAD 1 setup and an averaging time of 30 minutes, we found a wind speed comparability of 0.340 m s$^{-1}$ for the period of the triple DL stare and 0.323 m s$^{-1}$ for the period of the step/stare mode, respectively. The bias values were -0.261 m s$^{-1}$ and -0.010 m s$^{-1}$, respectively. The respective values for the VAD 2 setup are 0.636 m s$^{-1}$ and 0.468 m s$^{-1}$ as comparability and 0.589 m s$^{-1}$ and 0.336 m s$^{-1}$ as bias values. Hence, for the height of 90 m above ground, those methods performed similarly well in comparison to the sonic anemometer, proofing their applicability.

We further compared the triple DL with the single DL setups at the six different heights of the step/stare mode, and we found that the agreement between the VAD measurements and the VT DL measurements decreases with a decreasing averaging time. Especially, the RMSD and bias values also show a clear tendency that the agreement between the triple and single DL setups decreases with increasing height. Additionally, we found that an increased zenith angle in the single DL VAD measurements leads to a decreasing agreement with the VT DL measurements, especially at higher altitudes.

In summary, the single DLs performed well for our location and atmospheric conditions, especially since the two VAD modes considered here are primarily suited to derive turbulence variables and wind gusts they also give reliable mean wind values as well. The triple DL setup is advantageous when accurate measurements at a specific height (e.g., hub height of a wind turbine) are the primary goal or in complex terrain where the homogeneity of the flow field across the diameter of the VAD scan circle cannot be assumed (e.g. in valleys or at mountain ridges, close to coast lines). We also investigated a potential influence of atmospheric conditions on the measurement quality but did not find the measurement accuracy to be dependent on atmospheric stability, time of day, wind direction, or horizontal wind speed. This finding, which contradicts previous virtual DL measurements in a large-eddy simulation environment, may be attributed to our comprehensive set of quality control procedures so that only high-quality data were retained for this analysis.

We developed a set of advanced filters that resulted in an improvement in the data quality. However, as a result, the agreement with the sonic anemometer did only improve slightly, which shows that the data quality was quite high before. The data quality improvement only becomes apparent when analyzing the SNR + 1 values combined with the corresponding radial velocity values.

A reduction of the averaging time from 30 minutes to 2 minutes usually leads to a larger random error due to fewer independent samples, and we found this effect also in our data, which showed an increase of the RMSD from 0.382 m s$^{-1}$ to 0.639 m s$^{-1}$ and



a reduction of the Pearson's r value from 0.992 to 0.955 for the stare mode when compared to the sonic anemometer. The
corresponding effect can be seen in the values for the step/stare mode deteriorating from 0.450 m s$^{-1}$ to 0.547 m s$^{-1}$ and from
0.981 to 0.970, respectively.

Our findings highlight the importance of evaluating the effects of averaging times and averaging volumes on the derived
operational data products. The validity of assumptions required to evaluate scan schedules (i.e., the homogeneity assumption
for each VAD scan) should always be investigated, and where possible, quantified and applied to automatic filtering. Especially
if more complex measurement site footprints are considered, like in urban areas or complex terrain, this step cannot be omitted.
Using only one DL reduces the effort necessary for the setup of the instruments and their alignments. However, it can lead to
a data quality loss, especially at higher altitudes, which must be weighed against the above-mentioned advantages when
developing DL designs. If a continuous profile measurement across the atmospheric boundary layer is desired, other RHI-
related scan modes should be chosen for the triple DL or a single DL should be used to perform a VAD scan or similar pattern.
Moreover, it generally should be considered that a triple DL setup means a higher initial cost and more potential error sources
during the setup phase of the instruments. For measurements in urban environments (Zeeman et al., 2022), additional
challenges are posed by finding suitable sites within a certain proximity to each other for a multi-instrument system and by
obtaining usage permissions from the respective stakeholders.

For further studies, it might also be of interest to compare the measurements of a single DL additionally with those of a double
DL setup. Such a setup might be a good compromise between reduced cost and high data quality. Besides that, comparing DLs
with a taller instrumented tower as the one we had available, could demonstrate the performance of VT setups also for greater
heights, it could provide further insides into the differences between single and multiple DL systems, and it could help to
improve the filtering criteria we used. Moreover, an evaluation of different single and multi DL scanning strategies could allow
the development of additional recommendations when it is better to use a single or a specific type of multi-DL setup.6 Code
availability

## 6 Code availability

The code used in this study will be made available upon reasonable request.

## 7 Data availability

The data used in this study will be made available upon reasonable request.

## 8 Executable research compendium

Not applicable.



## 9 Sample availability

Not applicable.

## 10 Video supplement

Not applicable.

## 11 Supplement link

Not applicable.

## 12 Team list

Not applicable.

## 13 Author contributions

K. Wolz and M. Mauder setup the experiments and carried them out. M. Mauder designed the setup for this field experiment. C. Holst provided important code segments and assistance to help K. Wolz develop the code and carry out the data processing. F. Beyrich had organized the FESST@MOL field experiment and proposed the lidar setup and measurement strategy. E. Päschke provided important advice on issues related to noisy DL data. K. Wolz prepared the initial draft for this manuscript.

C. Holst, F. Beyrich, E. Päschke, and M. Mauder participated in the discussion of the intercomparison results and provided valuable contributions to the final manuscript version. M. Mauder supervised the work and provided help whenever needed.

## 14 Competing interests

The authors declare that they have no conflict of interest.

## 15 Disclaimer

Not applicable.



## 16 Acknowledgements

This work has been funded by the German Federal Ministry for Digital and Transport (BMDV) within the framework of the DWD program for extramural research, project: Quality Assessment of ground-based Lidar measurements in the Boundary layer (QALiBo): evaluation and verification of scanning strategies, quality tests and uncertainty quantification / contract number: 4819EMF05. Thanks go to C. Detring, R. Leinweber, and M. Kayser for operating the DWD DL systems and for performing the operational data processing.

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
