# Peer review of "Verifying triple and single Doppler lidar wind measurements with sonic anemometer data based on a new filter strategy for virtual tower measurements"

_EGUsphere, 2023_

## Author Comment (AC1)

Black text:     Referee comment. Line numbers relate to the original submission.

Red text:       Our response. Line numbers refer to revision.
* * *
**RC1:**

The authors conducted an observational validation and statistical comparison study between various Doppler lidar scanning strategies within the planetary boundary layer during a northern Germany field campaign in Summer 2020. Notably, triple and single Doppler lidar setups are compared against a tower-mounted sonic anemometer to determine the relative accuracy of each type of measurements at different heights, while a new filtering method is introduced for the virtual tower scanning strategy performed by the triple lidar setup. The rigorous statistical analyses provide a detailed understanding of the lidar measurements conducted and the efficacy of the filtering technique.

Nevertheless, the manuscript has grammatical errors throughout that should be addressed before the next submission. And probably more importantly, the uniqueness and new contributions for the field of lidar measurements and the usefulness of the new filtering method should be written more succinctly and emphasized more often. With these overall changes in mind, I return the paper to the authors for major revisions.

**Specific Comments**

*Introduction*: I think that the uniqueness of your triple DL setup and VT scan mode should be placed better in the context of past studies. How is your setup different? How are your objectives different than past work? This will allow the reader to better understand the motivation for the DL setup and scanning methods, and provide a basis for the reader to decipher how the Results shown later are useful to the community.

Thanks a lot for your valuable comments. We added further information on the past studies, hopefully helping to point out the differences to our experiments. Moreover, we added further information on our experiments to show their uniqueness. We changed the paragraph about the past studies. The updated paragraph about our objective is linked after your next comment.

*Starting with Line 98:* I think the sentences starting with "Our objective…" should be a separate paragraph since these sentences clearly outline your paper's objectives, and having them be a separate paragraph will draw more attention to them. Perhaps consider rephrasing some of these sentences as 'Research Questions' that are numbered and formulating your discussion in the Results section around the Research Questions.

We did create a separate paragraph and further changed parts of the second half of the introduction according to your comments. The new rewritten paragraphs are as follows:

[revised manuscript text omitted]

*Tables 2, 3, 4, 5*: I think it would be more beneficial to show the "Number of Observed Periods" instead as a percentage of the total number of possible observed periods in order to better emphasize the amount of data removed when applying a certain filter.

That is a good idea, we changed it accordingly in those tables. To make the tables more consistent we also changed that in Table 6 and Table 7.

Given that you have 7 Tables in this manuscript, I wondered if it could be possible to change one or multiple to a figure? Perhaps a box and whisker plot or a bar plot with error bars would be possible? I feel that it would perhaps be easier to digest the comparisons if less are in tabular format.

Thanks for your suggestion, we agree that seven is a high number of tables for this manuscript. Nevertheless, we think that tables represent our data best and, therefore, decided to keep them that way.

*Lines 497-499*: I would change the end of this sentence to indicate that 'the VT wind measurements perform equally well to wind measurements from the VAD DLs' since I think you want to show that using the VT method is a reputable method, correct?

After reconsidering this sentence, we concluded to omit it completely since its formulation was unclear and discuss about that topic directly in the next paragraph.

*Lines 516-518*: Sentences like this make it more difficult to understand the objectives of the paper. Are you trying to show that the VT method is better and therefore should be used more widely in the community? Are you trying to indicate that a single DL can be used instead of a triple DL setup? If your answer is no to these questions, then this sentence would seem to undercut your hypotheses.

Yes, we are trying to say that the VT methods works well but also that a single DL is still a viable method.

*Line 553*: Change "proofing" to "proving" if you would still like to keep this word. However, I would additionally suggest being careful from using this more absolute language. Using words like "indicating" or "reinforcing" I think tends to be safer since as scientists it is quite difficult for us to definitively show that a certain method is wholly justified.

You are right, thanks! We changed "proofing" into "demonstrating".

*Lines 568-569*: These two sentences seem to undermine the new filtering method you have outlined in this study. If the advanced filters do not improve the resulting comparison with the sonic anemometer data, would that not suggest that your filtering method should be altered? Or that the older filtering methods are still adequate to use?

We thoroughly tested out the filtering methods in many different ways and then decided on the ones we used. The data quality of even the unfiltered data was so high at the low altitude of 90 m above ground where the sonic anemometer is located that there were no relevant differences in data quality using the data we had available for this study. Therefore, we decided to include this sentence to give others the possibility to further test the filters. To make that clearer we added the following sentence behind the lines mentioned in your comment: "Therefore, further testing of the data filters on data collected at higher altitudes or in challenging meteorological situations should further demonstrate the filters' effectiveness."

**Technical Comments**

We would like to thank the reviewer for the detailed technical comments. We also did minor changes to different sentences throughout the script where we found other grammatical mistakes. We also changed some of the table descriptions accordingly.

I would suggest indenting each paragraph or placing a blank line between paragraphs to improve readability throughout.

Thanks for this good idea, we added a blank line between each paragraph.

*Lines 23 and 25*: "average time" and "averaging time" are both used in the same sentence position. However, for consistency, only one of these should be chosen.

We changed it to "averaging time".

*Lines 28-29*: "We found, that…" should be changed to "We found that".

Done.

*Lines 43-46*: "…can contribute to further increase the quality of the weather forecast…" could be written more succinctly. Perhaps "…can contribute to further increases in weather forecast quality…".

Done.

*Line 53*: Change "To determine wind speed and direction from a single DL a set of different scanning methods are well-established…" to something like "There are a set of well-established scanning methods to determine wind speed and direction from a single DL".

Done.

*Line 57*: I think it is generally not good practice to start a sentence with "that". I would suggest changing to something like "Alternatively, the 3-D winds can be obtained using different scanning methods like…".

Done, we changed this part of the sentence into: "Alternatively, the three-dimensional wind components can be obtained using different scanning methods like…".

*Line 70*: Change "especially, when" to "especially when".

Done.

*Line 84*: Change "respective DL they found" to "respective DL, they found".

Done.

*Line 117*: "number" can be removed to be more concise.

Done.

*Line 118*: Change "study," to "study:".

Done.

*Line 153*: Change "…for a comparison with the sonic anemometer" to "for comparison with the sonic anemometer".

Done.

*Line 161*: Change "Both these scan modes" to "Both of these scan modes".

Done.

*Line 181*: I would suggest identifying (1) as "Equation (1)" when used in the text. Also, if possible, I think centering the equations horizontally would be more correct formatting to use.

Done, we added "Equation" in front of the formula numbers when used in the text and centered the equations more horizontally.

*Line 202*: Change "In Figure 3" to "In Figure 3,".

Done.

*Figure 3*: Change "Overview over the meteorological conditions" to "Overview of the meteorological conditions".

Done.

*Line 216*: Change "from the north" to "from north".

Done.

*Line 220*: Change from "measurements" to "measurements,".

Done.

*Line 223*: Change "assure" to "ensure".

Done.

*Line 231*: Change "(see Table 2)" to "(see Table 2),".

Done.

*Line 232*: Change "…filter which…" to "…filter, which…".

Done.

*Line 233*: Change "…30 % that…" to "…30%, that…".

Done.

*Line 244*: Change "30 minutes averaging interval" to "30-minute averaging interval".

Done.

*Lines 267-268*: Change "When applying the MAD filter to constantly moving scanning strategies an increase" to "When applying the MAD filter to

constantly moving scanning strategies, an increase".

Done.

*Lines 268-269*: Change "If the actual radial velocity is around 0 ms$^{-1}$ the filter" to "If the actual radial velocity is around 0 ms$^{-1}$, the filter".

Done.

*Table 2*: Change "Person's" to "Pearson's".

Done.

*Line 282*: Change "all the filters" to "all of the filters".

Done.

*Line 293*: Change "as in the study of" to "as in".

Done.

*Lines 295-296 (similarly for Lines 297 and 300)*: I believe that standard figure citations should be "Figure 5a" and "Figure 5b" in this sentence.

Done, we changed that in all of the mentioned lines.

*Line 300*: Change "To" to "to".

Done.

*Figure 6, right y-axis label*: Change "m s$^{-1}$" to "(m s$^{-1}$)".

Done.

*Line 328*: Change "sonic anemometers" to "sonic anemometer".

Done.

*Line 405*: Change "time of the day" to "diurnal cycle".

Done.

*Lines 406-407*: This sentence is probably not necessary. I suggest simply mentioning "sonic anemometer" in the previous sentence.

Done, we removed the sentence and changed the previous one to We evaluate a potential influence of the diurnal cycle, the atmospheric stability expressed by the Obukhov-length $L$, the wind direction, and the mean wind speed measured by the sonic anemometer.

*Lines 411 and 524*: Should "sinus-wave-like" be "sine-wave-like"? I understand that sinus is likely derived from "sinusoidal", but I believe that the most typical usage is "sine" instead of "sinus".

Done.

*Lines 414-415*: I do not think that this sentence is really necessary. I would expect that a reader would be able to understand that the x-axis between two plots that show different variables may have a different extent.

Done, we removed the sentence.

*Line 438*: Change "comparisons this" to "comparisons, this".

Done.

*Line 445*: Change "stability we" to "stability, we".

Done.

*Line 452*: Change "We had to choose relatively relaxed filter criteria" to "We had to choose relatively relaxed filtering criteria".

Done.

*Lines 459-460*: I think that "as an example day" can be deleted.

Done.

*Line 471*: 'since only one instrument must be operated' can be deleted for succinctness.

Done.

*Lines 471-474*: 'Moreover, the setup of the instruments presents another error source when using multiple DLs compared to a single instrument, since the north orientation has to be carefully verified for all three systems. Hence, single DLs are certainly easier to handle and more cost-efficient than using a three DL setup' can be deleted. In my opinion, this information should be fairly intuitive to the reader.

Done.

*Line 484*: Change '…VT mode another measurement regime' to '…VT mode, another measurement regime…'.

Done.

*Lines 513*: Reword this part "based on the with height increasing spatial averaging area for the VAD measurements" to be grammatically correct and add a comma at the end of this phrase.

Done, we reworded the part into "…where we considered that the spatial averaging area increases with height for the VAD measurements,…".

*Lines 513-514*: Reword "the VT measurements present the actual state of the wind field" to be grammatically correct.

Done, we rephrased that part into "…convinced that the VT wind data represent the true wind at the VT location and are, therefore, more trustworthy at higher altitudes.".

*Lines 560-561*: Adding "they also give reliable mean wind values as well" onto the end of this sentence seems to make it a run-on sentence. I would suggest removing this phrase.

Done.

*Lines 583-584*: I would suggest making this portion more succinct – "other RHI-related scan modes should be chosen for the triple DL or a single DL should be used to perform a VAD scan or similar pattern".

Done, we changed that part of the sentence into ", other RHI-related scan modes for the triple DL or a single DL to perform a VAD or similar scan pattern should be chosen."

*Line 592*: Change "further insides" to "further insights".

Done.

*Line 593*: Change "multi" to "multiple".

Done.

*Lines 594-595*: Remove "6 Code availability" from the end of this paragraph

Done.

**RC 2:**

The authors present data from a field campaign in which triple-Doppler lidar measurements wind measurements were compared with those from a sonic anemometer on a 90-m tower and standard velocity-azimuth display (VAD) wind profile measurements from two different lidars operating in separate modes. Robust statistical analysis is presented, particularly of the triple-Doppler virtual tower measurements, showing the quality of the data. Additionally, the authors present a new data filtering methodology.

While the results appear to be scientifically sound, there are concerns over the uniqueness and significance of the study. There have been numerous past similar studies evaluating wind measurements from single and multi-Doppler lidars that the authors briefly review in the introduction. However, the authors need to do a better job of integrating their findings with respect to the literature and previous findings throughout the entire paper. Additionally, the authors should state up front at the end of the introduction what is unique about their analysis compared to the previous studies, as it appears very similar. As such, I recommend this paper be reconsidered after major revision in which the above general and following specific comments, building upon other reviewers' concerns, are addressed.

We rephrased the introduction with respect to the comments of reviewer 1 to better show what makes our study unique and to also further include the literature. In the final paragraph of the discussion we expanded the discussion to further include the literature.

At the beginning of the conclusions we added the sentence: "We also compared the measurements of the two VAD DLs with each other and with the sonic anemometer."

Specific Comments

*Figure 2:* It would be extremely helpful to add a scale to the figure in the lower left or right to show distance.

Done.

*Line 135:* I appreciate the methods used here to assess the azimuth angle with respect to true north as a robust way to determine the heading. Was a similar approach scanning up/down hard targets used as well to determine the pitch/roll of the lidar to assure it was level so that the elevation angle is known precisely (particularly over the virtual tower)?

To level the system, we used the pitch and roll sensors that are integrated in the instrument. We added the following sentence to that paragraph to clarify this: "For leveling the instruments, we used the system-integrated pitch and roll sensors which have a resolution of 0.1 °".

*Line 203*: The data used here covers nearly two months; it would be helpful to show the precipitation accumulation or rate as well. Furthermore, was data excluded for time periods when there were precipitation? During these time periods, the data may appear ok but the DL is really measuring the hydrometeor instead of aerosol movement, which is quite different than the air speed (particularly the vertical fall rate). Strongly suggest removing any periods of precipitation from subsequent analysis.

Thanks for your suggestion. We added the amount of precipitation that occurred during the period of the DL measurements to Figure 3. Before, we did not exclude the time periods where precipitation occurred. We tested manually removing the periods of rainfall now, however, the results stayed the same suggesting that the data filters handled those periods well. We added the following sentence directly above the updated Figure: "During the measurement campaign the precipitation was little with periods of low rainfall and a few major precipitation events." and this second sentence in chapter 2.5 Data filtering after Table 3: "We also tested excluding periods of rainfall from the analysis, there however, was no difference in the comparisons, suggesting that the data filters handled those periods well."

*Line 215:* '… western side of the lattice tower and pointing towards 190 deg'. Is this correct? If the sonic were on the western side, I'd expect it to be around 270 deg, as 190 would be on the southern side if I'm interpreting this correctly. Should it be the 'southern side'?

Yes, this is correct. The Falkenberg tower has a quadratic cross-section, and the booms are mounted on three sides of the tower lattice construction to be moved laterally for installation. Hence, the boom mounted on the western side of the lattice construction can be shifted roughly towards South or North. We added the word "laterally" to the text which hopefully helps to avoid mis-interpretation.

*Line 249:* What is the cause of the erroneous radial velocity values around 0 m/s? Is it known, such as due to hard target hits?

The erroneous radial velocity estimates around 0 m/s are a special type of non-white noise that seems to be quite common with the "Streamline" Doppler lidar systems, we found this type of noise in the data of several of these instruments. Päschke and Detring (2023, https://doi.org/10.5194/amt-2023-153) describe this in detail and suggest filtering methods to overcome this problem.

*Line 271:* Given the various stages of the data quality editing and filtering (e.g., SNR threshold, dip statistic, MAD, different time windows), it would be helpful to add a block diagram showing the flow of the different steps as a new figure.

Thanks for that good suggestion, we created a figure and added it as Figure 5. We also added the following additional sentence: "An overview of the different filtering and data processing steps can be seen in Figure 5."

*Line 304:* Could you add a brief description here of why the radial velocity values from the DL in Figure 5 (both panels) appear to have vertical lines of a large number of points at around 1, 3, and 5 m/s? This looks unrealistic, but maybe it's some artifact of underlying shear in the time period presented.

Possibly this is caused by the fact that the figure shows data from a full day and that the wind speed changed accordingly during the time frame. We took a second look at the wind data from that day which gave further confidence for that hypothesis.

*Line 313:* Could you succinctly explain why the VAD coverage is so poor at night? This is unexpected given the higher coverage from the VT, thus aerosols are present at higher altitudes and there are no clouds attenuating the signal. Is the poor coverage an artifact of the different filtering for the VAD measurements compared to the step/stare, due to the smaller number of pulses/rays, or something else? In general, throughout the entire paper, I feel like the discussion of the VAD results is lacking and insufficient consideration is given to fundamental differences in the lidar scanning parameters (zenith angle, pulse accumulation, focus length, etc.) with regard to explaining the results in the entire study.

Yes, the reduced data availability of the VAD measurements is probably caused by the reduced number of pulses per ray. We write that in Line 476/477 of the original manuscript.

In the discussion of the VAD results we added the following sentence: "Also, VAD 1 was operated with a focus of 500 m; whereas, the focus value of VAD 2 was set to 2000 m. Choosing a lower value might also improve the sensitivity of the measurements at the lower altitudes leading to a better agreement with the sonic anemometer data."

*Figures 8/9:* Recommend combining these two figures together with different coloring for what is currently in each figure to differentiate the two. For example, overlay the data of Figure 9 on current Figure 8, changing the colors of the data to red or some other color to differentiate the two. This will make it easier to visually see the differences.

Thanks for your suggestion, we decided not to overlay the two figures because this would result in many overlapping data points. However, we decided to combine the two figures by creating one figure with eight subplots where the old figure 8 is on the left side and the old figure 9 on the right. We feel, that this helps to make it easier to visualize the data that way.

*Lines 412-415 and Figure 10*: Suggest removing the figure and discussion as it does not add any value to what is presented. It is already clear that there is more disagreement between VT and VAD 1 at higher altitudes from previous data in the table, and data in Figure 10 and can be largely gleaned from what's in Figures 8/9.

We agree and did so.

*Lines 461*: Are these higher radial velocities associated possibly with second trip echoes or possibly contamination of clouds or other artifacts in the background noise (see Manninen et al 2016)?

That is plausible, we added the following sentence at that location: "These radial velocity values might possibly be related to second trip echoes, a contamination of clouds, or other artifacts in the background noise (Manninen et al. 2016)."

*Line 479*: Here, should specifically call out increasing the number of pulses/ray for additional integrations to increase the range of the DLs operating in VAD.

Thanks, we changed the sentence into: "Choosing another VAD configuration for the single DLs should further improve their performance, especially increasing the number of pulses per ray to potentially increase the range of the VAD measurements."

*Line 589*: The current dataset from this experiment would actually be conducive to analyzing a double-DL setup as well. Simply remove the vertically pointing lidar from the analysis and it could be easily done.

You are absolutely right, this is true. We, therefore, decided to remove the sentence and change the beginning of the paragraph slightly into: "For further studies, it might be of interest to compare DLs with a taller instrumented tower as the one we had available which could demonstrate the performance of VT setups also for greater heights, it could also provide further insights into the differences between single and multiple DL systems, and it could help to improve the filtering criteria we used."

*Line 595*: The words '6. Code availability' should be removed here as the section is below it.

Done.

References

Manninen, A.J., O'Connor, E.J., Vakkari, V. and Petäjä, T., 2016. A generalised background correction algorithm for a Halo Doppler lidar and its application to data from Finland. *Atmospheric Measurement Techniques*, *9*(2), pp.817-827.

**CC1:**

The authors conducted a field campaign comparing various Doppler lidar scanning strategies for measuring winds (speed, direction) within the planetary boundary layer.

The paper highlights the competence and the scientific soundness of the authors in this field.

However, I suggest to select a paper title which better reflects the main idea of the work and/or the new contribution in the field.

The word "Verifying"  may be definitely replaced by "comparing" at least.

According to my understanding of the paper content, I may suggest the following title:

Assessing the single Doppler lidar method for reliable/accurate wind speed measurement within the planetary boundary layer.

Thanks for your valuable comment on our paper, after further reconsideration about the title we decided to replace "verifying" with "comparing" as you suggested. However, we think that besides that the title represents good what we want to tell with our paper.

A new title may help the authors to give a common thread to the paper, and may inspire some modifications to the paper structure accordingly.
As for the first reviewer, I then suggest major revisions.

Most typesetting errors have been already listed by the first reviewer.